# Humoral and cellular immune responses against SARS-CoV-2 post-vaccination in immunocompetent and immunocompromised cancer populations

Elizabeth Titova,[1] Veronica W. Kan,[1] Tara Lozy,[1] Andrew Ip,[2,3] Kileen Shier,[4] Vittal P. Prakash,[4] Meghan Starolis,[4] Sara Ansari,[4] Kira Goldgirsh,[1] Seoyeon Kim,[1] Michael C. Pelliccia,[2,3] Aamirah Mccutchen,[1,3] Martinus Megalla,[1,3] Thomas S. Gunning,[2,3] Harvey W. Kaufman,[4] William A. Meyer III,[4] David S. Perlin[1,5]

**ABSTRACT**  Cancer patients are at risk for severe coronavirus disease 2019 (COVID-19) outcomes due to impaired immune responses. However, the immunogenicity of severe acute respiratory syndrome coronavirus 2 (SARS-CoV-2) vaccination is inadequately characterized in this population. We hypothesized that cancer vs non-cancer individuals would mount less robust humoral and/or cellular vaccine-induced immune SARS-CoV-2 responses. Receptor binding domain (RBD) and SARS-CoV-2 spike protein antibody levels and T-cell responses were assessed in immunocompetent individuals with no underlying disorders ($n = 479$) and immunocompromised individuals ($n = 115$). All 594 individuals were vaccinated and of varying COVID-19 statuses (i.e., not known to have been infected, previously infected, or "Long-COVID"). Among immunocompromised individuals, 59% ($n = 68$) had an underlying hematologic malignancy; of those, 46% ($n = 31$) of individuals received cancer treatment <30 days prior to study blood collection. Ninety-eight percentage ($n = 469$) of immunocompetent and 81% ($n = 93$) of immunocompromised individuals had elevated RBD antibody titers (>1,000 U/mL), and of these, 60% ($n = 281$) and 44% ($n = 41$), respectively, also had elevated T-cell responses. Composite T-cell responses were higher in individuals previously infected with SARS-CoV-2 or those diagnosed with Long-COVID compared to uninfected individuals. T-cell responses varied between immunocompetent vs carcinoma ($n = 12$) cohorts ($P < 0.01$) but not in immunocompetent vs hematologic malignancy cohorts. Most SARS-CoV-2 vaccinated individuals mounted robust cellular and/or humoral responses, though higher immunogenicity was observed among the immunocompetent compared to cancer populations. The study suggests B-cell targeted therapies suppress antibody responses, but not T-cell responses, to SARS-CoV-2 vaccination. Thus, vaccination continues to be an effective way to induce humoral and cellular immune responses as a likely key preventive measure against infection and/or subsequent more severe adverse outcomes.

**IMPORTANCE**  The study was prompted by a desire to better assess the immune status of patients among our cancer host cohort, one of the largest in the New York metropolitan region. Hackensack Meridian Health is the largest healthcare system in New Jersey and cared for more than 75,000 coronavirus disease 2019 patients in its hospitals. The John Theurer Cancer Center sees more than 35,000 new cancer patients a year and performs more than 500 hematopoietic stem cell transplants. As a result, the work was undertaken to assess the effectiveness of vaccination in inducing humoral and cellular responses within this demographic.

**KEYWORDS**  SARS-CoV-2, immune response, post-vaccination, immunocompromised, cancer

Address correspondence to David S. Perlin, David.Perlin@hmh-cdi.org.

Quest Diagnostics authors are employees of Quest Diagnostics and own stock in Quest Diagnostics.

10.1128/spectrum.02050-23 **1**

Nearly 2 million people in the United States are diagnosed with cancer annually and can be at increased risk of severe acute respiratory syndrome coronavirus 2 (SARS-CoV-2) infections and the associated adverse consequences due to suppressed immune responses (1). Immunosuppression among cancer patients is attributed to the cancers themselves and the associated cancer therapies (2). Immunosuppression is a risk factor for the development of various respiratory infections, given such affected individuals are less likely to naturally mount protective immune responses compared to immunocompetent individuals. While immunocompromised individuals account for approximately 2.7% of the general population, they account for 12.2% of hospitalized SARS-CoV-2 infected patients (3).

Some immunocompromised patients have lymphocytopenia with loss of up to 80% of peripheral T-cells and concurrent proliferation of $CD8^+$ T-cells (4). Following vaccination, development of robust neutralizing antibody responses in immunocompromised individuals would be diminished in comparison to immunocompetent individuals. Galmiche and colleagues found the risk of low immunogenicity of SARS-CoV-2 vaccines in immunocompromised populations, especially among patients who received solid organ transplants ($n = 5974$) and patients with hematological malignancy ($n = 7835$) relative to that observed in healthy controls (5). There is a lack of extensive research into the potential neutralizing role of humoral and cellular immune responses following vaccination in this vulnerable population, which formed the basis for this study.

Both vaccination and natural infection have been shown to trigger the development of neutralizing antibodies and reactive T-cells, with each serving complementary roles in the immune response to SARS-CoV-2 (6–11). Many questions remain regarding the durability of immunity conferred by either vaccination or natural infection, especially as time elapses following vaccination and as novel SARS-CoV-2 variants emerge.

Serologic testing can provide an objective measure of humoral response to either SARS-CoV-2 vaccination or infection. The vaccines available in the United States target the receptor binding domain (RBD) on the spike protein of SARS-CoV-2. Most individuals develop immunogenic responses to vaccinations, thus producing neutralizing antibodies that prevent infection by inhibiting the RBD from binding to ACE2 receptors on host cells (12, 13). Levels of RBD antibodies are correlated with viral neutralization and protection against severe disease (14–16). While antibody levels are an important correlate of protective immunity (17), they act in concert with antigen-reactive T-cells, which are a vital element of the long-term cellular immune response to SARS-CoV-2 (9, 18). The SARS-CoV-2 Omicron (B.1.1.529) variant has been observed to extensively evade neutralizing antibodies conferred by vaccination. Yet, cellular immunity induced by SARS-CoV-2 vaccination appears highly conserved (19, 20) and is expected to contribute to protection from reinfection and/or subsequent disease progression, even against Omicron variants. In recent years, interferon gamma (IFN-γ) release assays have been successfully used to test T-cell responses to stimulating antigens and may have the potential to assess lasting immunity in patients who have been infected with or vaccinated against SARS-CoV-2 (21).

Numerous studies have examined the prevalence and durability of SARS-CoV-2 antibodies as well as the T-cell response of individuals with prior SARS-CoV-2 vaccination or infection (22, 23). Immunocompromised patients, especially those with cancer, pose a challenge as they have displayed lower SARS-CoV-2 antibody seropositivity after vaccination with corresponding lower neutralization titers (24). Individuals with cancer, particularly those undergoing systemic anticancer treatments, B-cell-directed therapies, or hematopoietic stem-cell transplants that result in immunosuppressive states, are more susceptible to SARS-CoV-2 infection and may be at increased risk of coronavirus disease 2019 (COVID-19) mortality (8, 25–28).

Studies have shown that patients with hematologic or lymphoid malignancies, including leukemias, myelodysplastic syndromes, myeloproliferative neoplasms, lymphomas, and multiple myeloma (24), suffer from increased mortality due to COVID-19. Vaccinated patients with such hematologic malignancies have an estimated

mortality rate of 12.4% within 30 days of COVID-19 infection onset (22). This vulnerability may be secondary to immune system defects that are a result of the cancer itself combined with lymphoma-directed therapies (18). A United Kingdom-based study showed that 52% of patients with lymphoma undergoing active cancer treatment did not have detectable humoral responses after receiving two SARS-CoV-2 vaccine doses. In addition, 60% of patients undergoing anti-CD20 therapy within 1 year of receiving anticancer treatment did not have detectable antibodies following two vaccine doses (29). Individuals with immunosuppression, including those with hematologic or lymphoid malignancies, can present with persistent COVID-19 symptomatology. The persistence of this disease has the potential to facilitate the emergence of novel SARS-CoV-2 variants due to high viral burdens and rapid viral evolution in immunocompromised patients (28, 30). Thus, there remains a need to assess and quantify neutralizing antibody titers and T-cell responses against SARS-CoV-2 in vaccinated, immunocompromised patients and if appropriate, suggest mitigation strategies to reduce the risk of subsequent adverse SARS-CoV-2 outcomes.

In this pilot study, we investigated threshold levels of SARS-CoV-2 protection and compared the baseline levels of humoral and cellular immunity in a cohort of immunocompromised patients, many with cancer or autoimmune disorders, as well as in immunocompetent individuals. We hypothesized that immunocompromised individuals would mount less robust humoral and cellular immune responses. This study examined how the presence of cancer and the receipt of anticancer therapies affected the humoral and cellular immune responses following vaccination or natural infection compared to immunocompetent individuals.

## MATERIALS AND METHODS

### Experimental design and participant recruitment

In this cross-sectional study, we aimed to compare humoral and cellular responses following administration of current U.S. Food and Drug Administration (FDA) emergency use authorization (EUA) approved SARS-CoV-2 vaccines and boosters (i.e., Moderna/Pfizer BioNTech mRNA vaccines/J&J) in individuals recruited from several clinical categories. Eligible participants, recruited from 14 February 2022 to 3 June 2022, included vaccinated individuals 18 years of age or older who provided informed consent. Individuals were recruited at Hackensack Meridian Health (HMH) network locations including the John Theurer Cancer Center in Hackensack, NJ, Jersey Shore University Medical Center in Neptune, NJ, and Center for Discovery and Innovation (CDI) in Nutley, NJ, as well as Quest Diagnostics locations in Clifton, NJ, and Chantilly, VA. Participants included HMH or Quest Diagnostics employees, HMH patients, and other volunteers.

All individuals were separated into cohorts based on any disease status that might affect their immune response to SARS-CoV-2. Participants were further separated based on infection status and post-acute disease sequelae (post-acute sequelae of SARS-CoV-2 infection [PASC]) diagnosis. Individuals who had been previously infected with SARS-CoV-2 and who exhibited clinical conditions consistent with CDC-defined PASC were defined as "Long-COVID" (27). Immunocompromised participants were individuals who had a self-reported autoimmune disorder or received chemotherapy, immunotherapy, and/or radiation treatments/medications. All other individuals were classified as immunocompetent. All participants self-reported their COVID-19 disease status at the time of blood collection. Individuals who reported no prior SARS-CoV-2 infection, but had detectable SARS-CoV-2 nucleocapsid (N) antibodies, were reclassified into the "previously infected" cohort during data analysis.

This study was approved by the Hackensack Meridian Health (HMH) Institutional Review Board (IRB) (Pro2020-0633, Pro2020-0414). Under IRB-approved protocols and after providing informed consent, all participants had peripheral blood specimens collected at a single time point> more than 12 days after receiving a SARS-CoV-2 vaccination or booster. Antibody and T-cell responses were measured using the HMH-CDI

research use only (RUO) SARS-CoV-2 RBD ELISA Assay and the QuantiFERON SARS-CoV-2 Starter & Extended Pack Test, respectively. All individuals were asked to self-report personal and clinical information pertinent to the study. Furthermore, the Roche Cobas Elecsys Anti-SARS-CoV-2 Nucleocapsid Assay was used to assess recent SARS-CoV-2 infection history in the study population and to identify individuals who were previously infected with SARS-CoV-2 but had self-reported no prior COVID-19 infection.

## SARS-CoV-2 antibody response

### HMH-CDI Research Use Only (RUO) SARS-CoV-2 RBD ELISA Assay

The COVID-19 enzyme-linked immunosorbent assay (ELISA) protocol was performed as previously published (29). Participant antibody reactivity was established by measuring optical density (OD) at 490 nm via TECAN Sunrise microplate reader with Magellan software. A baseline sample value was set at $OD_{490} = 0.11$, and area under the curve (AUC) was calculated. AUC values below 1.00 were assigned a value of 0.50 for graphing and calculation purposes. Distribution of the different antibody isotypes in specimens that reacted with another ELISA was established with the use of different secondary antibodies, including anti-human IgA (α-chain-specific) horseradish peroxidase (HRP) antibody (Sigma A0295) (1:3,000), anti-human IgM (µ-chain-specific) HRP antibody (Sigma A6907) (1:3,000), anti-human IgG1 Fc-HRP (Southern Biotech 262 9054-05) (1:3,000), anti-human IgG3hinge-HRP (Southern Biotech 9210-05) (1:3,000), and anti-human IgG4 Fc-HRP (Southern Biotech 9200-05). Data were analyzed in Prism 7 (GraphPad).

A high antibody value was defined as a detectable result greater than 1,000 U/mL within the HMH-CDI RUO SARS-CoV-2 RBD ELISA Assay (29). IgG antibody responses were categorized as follows: below the level of quantification, 100–499, 500–999, 1,000–9,999, 10,000–50,000, and >50,000 U/mL.

## Nucleocapsid SARS-CoV-2 antibodies

The U.S. FDA EUA approved Roche Cobas Elecsys Anti-SARS-CoV-2 immunoassay qualitatively detects SARS-CoV-2 N antibodies, using a recombinant protein representing the N antigen (26). The assay consists of two incubation periods that result in a sandwich complex of biotinylated SARS-CoV-2-specific recombinant antigen and SARS-CoV-2-specific recombinant antigen labeled with ruthenium complex. Microparticles from the reaction mixture were magnetically bound to the surface of an electrode. When a voltage was applied to the electrode, a chemiluminescent emission was induced. Results were determined by the photomultiplier software, and the electrochemiluminescence signal obtained from the reaction product of the sample was compared to the cutoff value obtained during calibration.

Results were interpreted as non-reactive by a cutoff index <1.0. Non-reactive results indicated specimen was negative for anti-SARS-CoV-2 negative. Reactive results, which were positive for anti-SARS-CoV-2 antibodies, had a ≥1.0.

## SARS-CoV-2 T-cell response

### QuantiFERON SARS-CoV-2 starter and extended pack test

T-cell responses were evaluated with the Qiagen QuantiFERON SARS-CoV-2 Starter Set Blood Collection Tubes, which consists of two antigen tubes, SARS-CoV-2 Ag1 and SARS-CoV-2 Ag2 (25). This kit contains a combination of antigens specific to SARS-CoV-2 to stimulate lymphocytes in heparinized whole blood involved in cell-mediated immunity, as well as an internal positive (mitogen value) and negative control. The mitogen value was used to determine overall immune function. SARS-CoV-2 Ag1 stimulates CD4[+] (helper T-cells) using epitopes derived from the S1 subunit (RBD) of the spike protein. SARS-CoV-2 Ag2 induces stimulation of CD4[+] and CD8[+] (killer T-cells) with epitopes from the S1 and S2 subunits of the spike protein. The QuantiFERON SARS-CoV-2

Extended Set Blood Collection Tubes consists of one antigen tube, SARS-CoV-2 Ag3, which stimulates CD4$^+$ and CD8$^+$ via epitopes from S1 and S2, plus immunodominant CD8$^+$ epitopes derived from the whole genome. Plasma from the stimulated samples was used for detection of IFN-γ using the QuantiFERON ELISA.

## Statistical analysis

Summary statistics included median and interquartile range (IQR) or mode and frequency for continuous variables respective of each underlying distribution. Categorical variables were represented by counts and associated frequencies. A composite T-cell response metric was calculated using a weighted sum of each individual antigen response. T-cell responses following individual antigen stimulation are displayed in Fig. S1. Weights were determined using principal component analysis. Group comparisons were performed using Spearman's correlation, one-way analysis of variance (ANOVA), or Welch's ANOVA (if heteroscedasticity was present) with a significance threshold of 0.05. Student's *t*-test and/or Games-Howell *ad hoc* analyses were utilized when significant differences were found, depending on the type of statistical analysis used. Pairwise mean comparisons were assessed using the Student's *t*-test. All statistical analyses were performed using JMP, version 17.0 (SAS Institute Inc., Cary, NC, 1989–2022).

## RESULTS

A total of 594 individuals were recruited (Table 1); 71.4% ($n = 424$) were females, and the overall median age was 51 years (IQR 38–62). Additionally, 65.3% ($n = 388$) of individuals were White, non-Hispanic, 23.9% ($n = 142$) were other non-Hispanic, and 10.8% ($n = 64$) were Hispanic or Latino. Participants had a mean of three SARS-CoV-2 vaccine doses.

Based upon detectable SARS-CoV-2 nucleocapsid antibodies, 18.8% (62/329) of immunocompetent individuals and 21.0% (17/81) of immunocompromised individuals who reported no prior SARS-CoV-2 infection were reclassified as "previously infected." Overall, 55.7% ($n = 331$) were categorized as uninfected, 33.0% ($n = 196$) as previously infected, and 11.3% ($n = 67$) as self-reporting with Long-COVID symptoms. Of the 115 immunocompromised participants recruited into this study, 35 had autoimmune disorders and 80 had cancer. Of the individuals with cancer, 68 had hematologic malignancies and 12 had carcinomas. Treatment within 30 days before their blood collection event was noted in 45.6% of individuals with hematologic malignancies and 83.3% of individuals with carcinomas.

The majority of both immunocompromised (75.7%) and immunocompetent (75.8%) patients exhibited a mitogen-stimulated IFN-γ response of >10 IU/mL, with no statistically significant difference ($P = 0.977$) observed between the two cohorts.

When comparing T-cell response levels with antibody levels in the immunocompetent population, a consistent positive trend was observed for both mean T-cell responses and antibody levels [rs(477) = 0.184, $P < 0.01$] (Fig. 1). No such pattern was observed in the immunocompromised population [$F(2, 90) = 0.4508$, $P = 0.64$], in which T-cell responses varied independent of antibody levels. 97.9% (469/479) of immunocompetent and 80.9% (93/115) of immunocompromised individuals had RBD antibody titers >1,000 U/mL, and among these individuals, 60% (282/469) and 44% (41/93) had elevated T-cell responses above the assay cutoff threshold, respectively. A robust outlier analysis using univariate (quantile ranges) and multivariate (k nearest neighbor) was performed but did not significantly impact the results when identified outliers were excluded from the analysis.

The mean T-cell composite responses of immunocompetent compared to immunocompromised cohorts were different ($P < 0.001$) (Fig. 2). Within the immunocompetent cohort, individuals with previous infection had a higher composite T-cell response than those who were uninfected ($P < 0.05$). In the immunocompromised cohort, the Long-COVID sub-cohort had a higher mean T-cell response than the uninfected ($P < 0.05$) or previously infected ($P < 0.01$) immunocompromised individuals.

**TABLE 1** Demographics of recruited participants stratified into cohorts based on immune and COVID-19 disease status

| Demographics | Immunocompetent (*n* = 479) | | | Immunocompromised (*n* = 115) | | | Total (*n* = 594) |
| --- | --- | --- | --- | --- | --- | --- | --- |
| | Uninfected (*n* = 267) | Previously infected (*n* = 160) | Long-COVID (*n* = 52) | Uninfected (*n* = 64) | Previously infected (*n* = 36) | Long-COVID (*n* = 15) | |
| Sex, *n* (%) | | | | | | | |
| Female | 203 (76.0) | 113 (70.6) | 46 (88.5) | 38 (59.4) | 15 (41.7) | 9 (60.0) | 424 (71.4) |
| Male | 64 (24.0) | 47 (29.4) | 6 (11.5) | 26 (40.6) | 21 (58.3) | 6 (40.0) | 170 (28.6) |
| Race/ethnicity, *n* (%) | | | | | | | |
| Non-Hispanic White | 183 (68.5) | 83 (51.9) | 36 (69.2) | 54 (84.4) | 20 (55.6) | 12 (80.0) | 388 (65.3) |
| Non-Hispanic other | 68 (25.5) | 49 (30.6) | 10 (19.2) | 6 (9.4) | 7 (19.4) | 2 (13.3) | 142 (23.9) |
| Hispanic or Latino | 16 (6.0) | 28 (17.5) | 6 (11.5) | 4 (6.3) | 9 (25.0) | 1 (6.7) | 64 (10.8) |
| Age, median (IQR) | 52 (39–62) | 44 (32–59) | 46 (31–56) | 65 (57–74) | 55 (39–70) | 54 (46–64) | 51 (38–62) |
| COVID vaccine doses (#), mode (%) | 3 (82.4) | 3 (74.4) | 3 (71.2) | 3 (85.9) | 3 (63.9) | 3 (73.3) | |

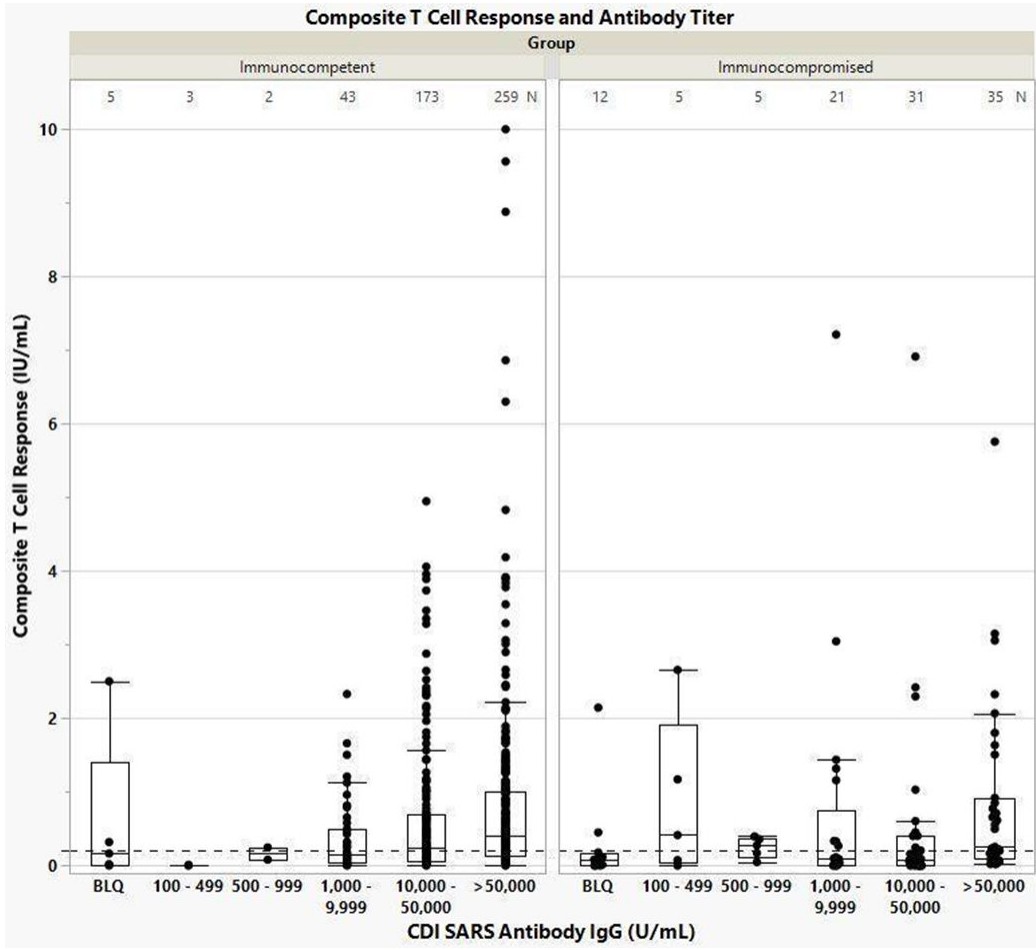

**FIG 1** Neutralizing antibody levels and composite T-cell responses following stimulation by Ag1, Ag2, and Ag3 in immunocompetent and immunocompromised populations. T-cell response positivity threshold of 0.2 IU/mL is indicated as a dashed line on the *y*-axis.

Based on Welch's ANOVA, a difference was observed between the mean T-cell composite response among different clinical cohorts due to unequal variances ($P < 0.001$) (Fig. 3). The statistically significant difference ($P < 0.001$) from Welch's ANOVA was

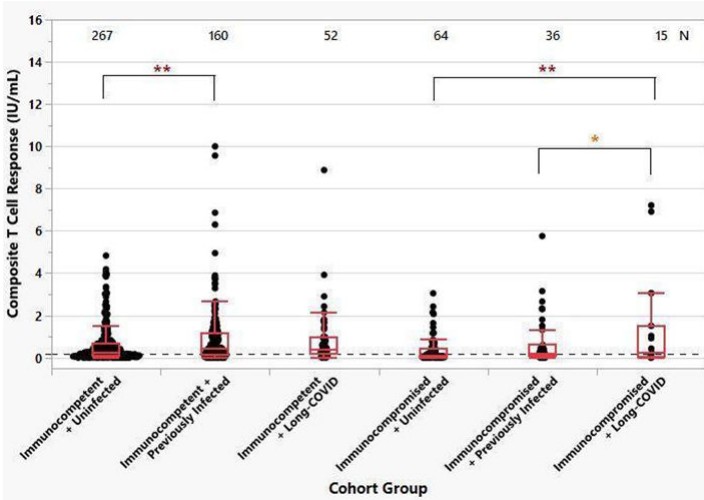

**FIG 2** Composite T-cell response of immunocompetent and immunocompromised patient populations. An ANOVA was run on composite scores, and a statistically significant difference ($P < 0.001$) was found. A pairwise $t$-test was performed to look for significant groups. *$P < 0.05$ and **$P < 0.01$.

primarily due to the mean difference between immunocompetent and carcinoma cohorts, based upon Games-Howell *post hoc* used for pairwise comparisons. While the variance in the immunocompetent cohort was large, there was little variance in the carcinoma cohort.

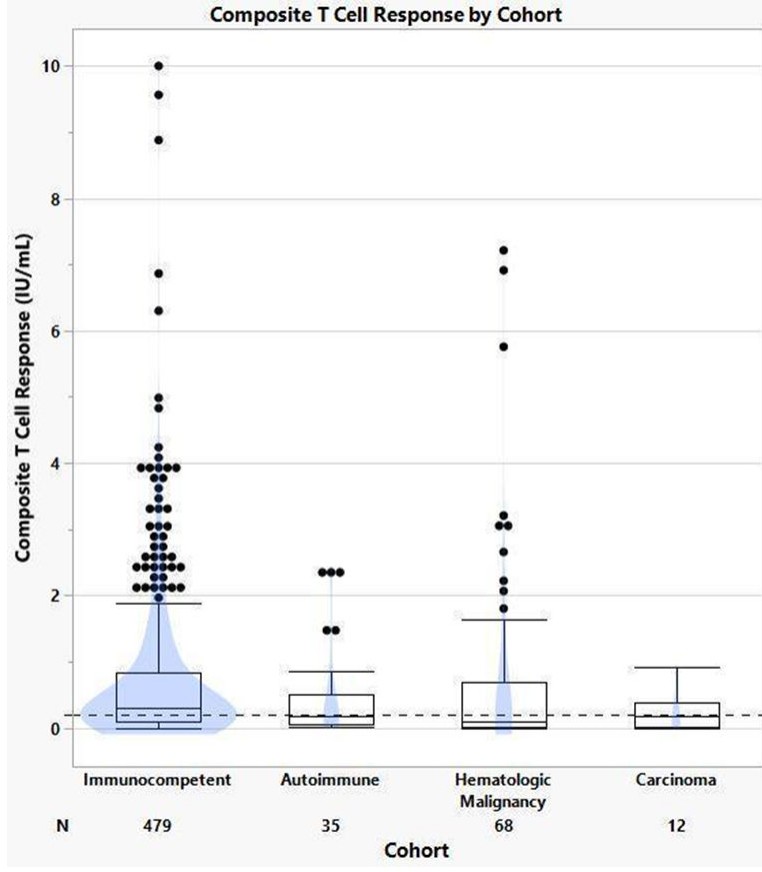

**FIG 3** Composite T-cell responses based on diagnosis cohorts.

There was no statistically significant difference found between the T-cell responses of the immunocompetent and hematologic malignancy cohorts ($P = 0.992$). The large variance among participant specimens of the hematologic malignancy group contributes to the wider confidence interval.

When RBD antibody responses were compared based on underlying disease (Fig. 4), the largest variance was seen in the hematologic malignancy cohort, followed by the carcinoma cohort. The autoimmune and immunocompetent cohorts showed mean antibody response >10,000 U/mL.

## DISCUSSION

This study found most SARS-CoV-2 vaccinated individuals mounted robust cellular and/or humoral responses, though higher immunogenicity was observed among the immunocompetent compared to immunocompromised populations, including cancer patients. Robust antibody responses correlated with elevated T-cell responses in the immunocompetent cohort. This trend likely reflects the biological function of CD4$^+$ cells in stimulating B-cells to produce antibodies (31). The immunocompromised cohort does not follow the same trend (Fig. 1), likely due to the effects of B-cell targeted agents in the treated immunocompromised patient cohorts. These agents include anti-CD20 monoclonal antibodies (i.e., rituximab, obinutuzumab) and targeted B-cell pathway inhibitors (i.e., BTK inhibitors, BCL-2 inhibitors, proteasome inhibitors). Such B-cell targeted therapies were associated with depleted antibody responses, yet we found that 43% (19/44) of immunocompromised patients undergoing these therapies still mounted a robust T-cell response. This is consistent with Rouhani and colleagues' study that analyzed 14 patients receiving B-cell-directed therapies (i.e., anti-CD20 antibodies, multiple myeloma therapies) and demonstrated decreased SARS-CoV-2 antibody titers (12).

In immunocompromised transplant recipients, natural infection with influenza viruses produces greater diversity in humoral responses compared to vaccination (3, 32). For SARS-CoV-2, enhanced neutralizing antibody responses are induced in healthy individuals by a single dose of BNT162b2 or after natural SARS-CoV-2 infection (33). Similarly, we found that in addition to antibody responses, natural infection triggers an elevated T-cell response in most vaccinated individuals.

Natural infection appears to elevate T-cell response in both immunocompetent and immunocompromised vaccinated populations (Fig. 2). A study by Reynolds and colleagues is consistent with our findings, demonstrating that individuals who only received one vaccine dose and had a prior infection showed enhanced T-cell immunity, unlike those who only had one dose but were previously uninfected (34).

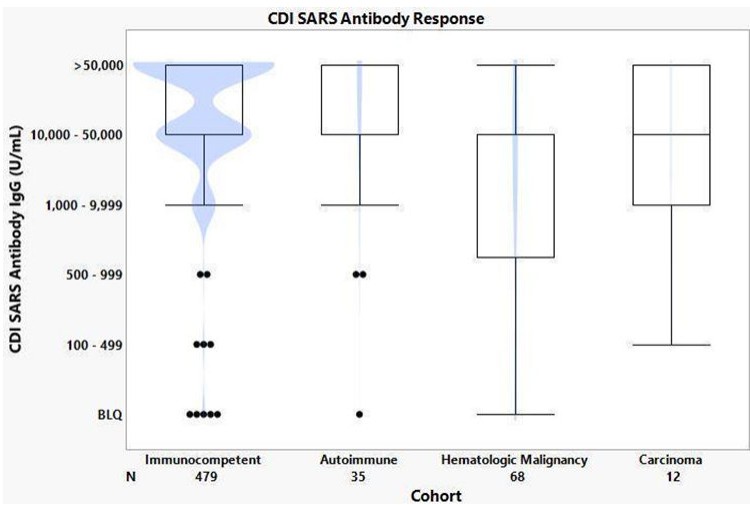

**FIG 4** Antibody responses based on diagnostic cohorts.

There was a significantly higher mean composite T-cell response in the immunocompetent cohort vs the carcinoma cohort (Fig. 3). This suggests that either carcinoma treatments or the disease itself impairs the T-cell response to SARS-CoV-2 exposure via natural infection or vaccination. However, a larger cohort study of individuals with solid tumors ($n = 96$) found that a majority of the patients developed T-cell responses with no significant difference compared to immunocompetent individuals following vaccination (25).

The varied T-cell response seen in the hematologic malignancy cohort (Fig. 3) was not statistically different when compared to the immunocompetent cohort. However, outliers seen in the hematologic malignancy cohort can be attributed to differences in the underlying type of hematological disease of individuals. Conditions including chronic lymphocytic leukemia and Hodgkin's lymphoma result in a dysregulated immune response; therefore, individuals with these disorders are likely to exhibit an abnormal immune response to vaccination (25, 35). Five of the 10 T-cell outliers in the hematologic malignancy cohort represent individuals with potential dysregulated T-cell conditions.

The antibody variance seen within the hematologic malignancy and carcinoma cohorts (Fig. 4) can be accounted for by the presence of B-cell suppressing drugs as well as the length of time since each individual's last B-cell suppressing treatment. This is consistent with other studies that indicate titer variation in patients being treated for solid tumors (5, 12).

Our data support the concept of a suppressive role of B-cell targeted therapies on antibody responses, but do not indicate any implications on parallel T-cell responses in those same individuals. Those receiving B-cell targeted therapies were present in both the hematologic malignancy and carcinoma cohorts. Within these immunocompromised sub-cohorts, widespread variance in antibody responses was observed and may be attributed to the length of time since the patient's last immunosuppressive treatment. As a result, vaccinated patients receiving B-cell targeted therapies seem to derive clinical benefit from administration of SARS-CoV-2 monoclonal or polyclonal antibody preparations to increase humoral immunity (12, 35, 36). The data suggest individuals undergoing B-cell targeted immunosuppression continue to maintain a cellular response against SARS-CoV-2 antigens.

Nineteen percentage (49/410) of participants who reported never having COVID-19 were reactive for N SARS-CoV-2 antibodies. This suggests that these participants had been unknowingly infected with SARS-CoV-2, which is consistent with earlier studies that reported approximately 40% of SARS-CoV-2 infected patients with similar exposure histories, and treatment were asymptomatic (37). Despite having overall lower viral burdens and shedding virus for a shorter time than symptomatic patients, asymptomatic persons are still able to transmit the virus, which emphasizes the need for continued SARS-CoV-2 surveillance (38).

While efforts were made to diversify the study population, there were noted disparities in sex, race, and ethnic populations that may have introduced confounding variables to the study. Variation in vaccine manufacturers and number of doses received, in diagnosis and treatment received, and in the length of elapsed time since last vaccine dose/immunosuppressive treatment may have influenced the observed results.

The limitations of this study include the absence of diagnosis, treatment, and SARS-CoV-2 infection status information. These factors should be further investigated to determine their possible influence on depleting T-cell responses. The small sample size ($n = 12$) but low variance in the T-cell responses of the carcinoma cohort supports the need for future studies that focus on recruiting such individuals to verify this observation. Lastly, further evaluation of individuals with conditions involving dysregulated immune responses and the resulting effects on elevating T-cell responses should be considered.

## Conclusion

Overall, this study demonstrates that most immunocompetent and immunocompromised SARS-CoV-2 vaccinated individuals mount a robust cellular and/or humoral

response. The immune response was stronger in immunocompetent than in immunocompromised individuals. This robust immune response may provide protection against future infection and subsequent risk of disease progression. While humoral immunity is a correlate of protection against SARS-CoV-2, cellular immunity evaluations may also need to be considered when assessing the immune status of vulnerable immunocompromised individuals, especially cancer patients undergoing treatment.

## ACKNOWLEDGMENTS

The authors would like to thank Quest Diagnostics for their financial support, participant recruitment, and laboratory testing. The authors appreciate the support from the staff at the John Theurer Cancer Center and Jersey Shore University Medical Center in participant recruitment. The authors would like to thank Kelly K. Yen and Erika Shor, Ph.D. from the Center for Discovery and Innovation for their extensive participation in the revision process.

## AUTHOR AFFILIATIONS

[1]Center for Discovery and Innovation, Hackensack Meridian Health, Nutley, New Jersey, USA
[2]John Theurer Cancer Center, Hackensack, New Jersey, USA
[3]Hackensack Meridian School of Medicine, Nutley, New Jersey, USA
[4]Quest Diagnostics, Secaucus, New Jersey, USA
[5]Georgetown Lombardi Comprehensive Cancer Center, Washington, DC, USA

## AUTHOR ORCIDs

Elizabeth Titova http://orcid.org/0000-0001-7136-7828
Tara Lozy http://orcid.org/0000-0002-6478-5690
Kileen Shier http://orcid.org/0000-0001-8646-5201
David S. Perlin http://orcid.org/0000-0002-1268-5524

## AUTHOR CONTRIBUTIONS

Elizabeth Titova, Conceptualization, Data curation, Formal analysis, Investigation, Project administration, Writing – original draft, Writing – review and editing | Veronica W. Kan, Investigation, Writing – original draft, Writing – review and editing | Tara Lozy, Data curation, Formal analysis, Methodology, Software, Validation, Writing – original draft, Writing – review and editing | Andrew Ip, Conceptualization, Investigation, Resources, Writing – review and editing | Kileen Shier, Conceptualization, Investigation, Methodology, Project administration, Resources, Writing – review and editing | Vittal P. Prakash, Investigation, Project administration, Resources, Writing – review and editing | Meghan Starolis, Conceptualization, Investigation, Supervision, Writing – review and editing | Sara Ansari, Investigation, Project administration, Resources, Writing – review and editing | Kira Goldgirsh, Project administration | Seoyeon Kim, Data curation, Investigation, Writing – original draft | Michael C. Pelliccia, Writing – original draft | Aamirah Mccutchen, Investigation, Project administration | Martinus Megalla, Writing – original draft | Thomas S. Gunning, Writing – original draft | Harvey W. Kaufman, Conceptualization, Formal analysis, Investigation, Project administration, Resources, Supervision, Writing – review and editing | William A. Meyer III, Conceptualization, Formal analysis, Funding acquisition, Investigation, Methodology, Project administration, Resources, Supervision, Writing – review and editing | David S. Perlin, Conceptualization, Formal analysis, Funding acquisition, Investigation, Methodology, Project administration, Resources, Supervision, Visualization, Writing – review and editing

## ETHICS APPROVAL

This study was approved by the Hackensack Meridian Health (HMH) Institutional Review Board (IRB) (Pro2020-0633, Pro2020-0414).

## ADDITIONAL FILES

The following material is available online.

### Supplemental Material

**Figure S1 (Spectrum02050-23-S0001.docx).** Neutralizing antibody levels and individual T-cell responses following stimulation by Ag1, Ag2, and Ag3 in immunocompetent and immunocompromised populations.

### Open Peer Review

**PEER REVIEW HISTORY (review-history.pdf).** An accounting of the reviewer comments and feedback.

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
