## [Reviewer comments · Microbiology Spectrum]

Microbiology Spectrum

Humoral and Cellular Immune Responses Against SARS-CoV-2 Post-Vaccination in Immunocompetent and Immunocompromised Cancer Populations

Elizabeth Titova, Veronica Kan, Tara Lozy, Andrew Ip, Kileen Shier, Vittal Ponraj, Meghan Starolis, Sara Ansari, Kira Goldgirsh, Seoyeon Kim, Michael Pelliccia, Aamirah Mccutchen, Martinus Megalla, Thomas Gunning, Harvey Kaufman, William Meyer III, and David Perlin

Corresponding Author(s): Elizabeth Titova, Hackensack Meridian Health

Review Timeline:

Submission Date:	May 17, 2023
Editorial Decision:	August 12, 2023
Revision Received:	December 3, 2023
Editorial Decision:	December 17, 2023
Revision Received:	December 31, 2023
Accepted:	January 4, 2024

Editor: Tulip Jhaveri

Reviewer(s): The reviewers have opted to remain anonymous.

Transaction Report:

DOI: <https://doi.org/10.1128/spectrum.02050-23>

August 12, 2023

Ms. Elizabeth Titova
Hackensack Meridian Health
Center for Discovery and Innovation
111 Ideation Way
Nutley, NJ 07110

Re: Spectrum02050-23 (Humoral and Cellular Immune Responses Against SARS-CoV-2 Post-Vaccination in Immunocompetent and Immunocompromised Cancer Populations)

Dear Ms. Elizabeth Titova:

Thank you for submitting your manuscript to Microbiology Spectrum. Extensive revisions are required before this manuscript can be considered for publication. When submitting the revised version of your paper, please provide (1) point-by-point responses to the issues raised by the reviewers as file type "Response to Reviewers," not in your cover letter, and (2) a PDF file that indicates the changes from the original submission (by highlighting or underlining the changes) as file type "Marked Up Manuscript - For Review Only". Please use this link to submit your revised manuscript - we strongly recommend that you submit your paper within the next 60 days or reach out to me. Detailed instructions on submitting your revised paper are below.

Link Not Available

Sincerely,

Tulip Jhaveri

Journals Department
Reviewer comments:

Reviewer #1 (Comments for the Author):

The point of departure of the authors is novel and interesting, they hypothesized that cancer vs non-cancer individuals would mount less robust humoral and/or cellular vaccine-induced immune SARS-CoV-2 responses and conducted the data collection and research. Totally, the draft was well-written, and the statics collected were clear. However, some detailed problems should have been concerned during the review process.

1. The format of the article should be corrected if possible, and no cover letter have I found in the submitted files. No "purpose, methods, etc." should be listed in the abstract, the authors could benefit from the published articles in "Microbiology Spectrum",

and the "Importance" of the article should be added if possible.

2. In the "Methods", the information the authors introduced should be more detailed. What's the basic evaluation criterion to distinguish the "immunocompetent individuals with no underlying disorders" and the "immunocompromised individuals"? In the purpose of the article, the author introduced that cancer patients are at risk for severe COVID-19 outcomes due to impaired immune responses, whether the "immunocompromised individuals" are cancer patients only or have other symptoms? Is there any criterion to distinguish them from healthy individuals? So, I think that's important to introduce more details in this part.
3. What kind of RBD antibodies were tested in this research? Omicron or other types? How are the accuracy and positive rate for testing out? Because the RBD is mutating, it may be better to clarify this.
4. The graphs shown in the draft were not clear enough, please submit the original graphs. Figure legends in the draft should better contain the statistical percentages and total tested numbers for easier understanding to readers if the authors intend to explain the results. Figure legend 1, the CDI SARS antibody IgG statement was not accurate enough, and I think it may be SARS-CoV-2.
5. It is better to include a statistical analysis of the relationship between the levels of RBD IgG antibodies in the immunocompetent and immunocompromised population in this research who was first infected and have determined the time of infection to explore the duration of antibody.
6. Within the immunocompetent cohort, individuals with previous infection had a higher composite T-cell response (What is the level?) than those who were uninfected ($P < 0.05$). In the immunocompromised cohort, the Long-COVID sub-cohort had a higher mean T-cell response (What is the level?) than the uninfected ($P < 0.05$) or previously infected ($P < 0.01$) immunocompromised individuals.
7. The statements for all Figures should be more clear and elucidate the significance of the results, not just some statistical illustration without a conclusion, that may weaken the significance of your research.
8. More statistical data should be added if possible, such as mentioned in question 5.

Reviewer #2 (Comments for the Author):

General comment

The study of Titova et al. analyzes SARS-CoV-2 specific antibody and T cell responses in immunocompetent and immunocompromised cohorts. Because the populations analyzed are subject to many variables, the study requires a comprehensive analysis of the influence of those variables. This is not adequately presented. The main findings that immunocompromise is associated with impaired antibody responses and B cell compromise is independent of T cell responsiveness is largely unsurprising, therefore a more detailed presentation of the groups by specific therapy and disease condition would have been of greater value.

Specific comments

1. Group sizes and gender distributions are disparate among groups. The authors recognize this issue in the Discussion but did not attempt to analyze groups by gender or age.
2. The authors refer to "robust" antibody and T cell responses, but do not define what is robust. This should be defined in terms of relationship to resistance to infection or time since antigen exposure. Since antibody levels rise and decay, the time since vaccination or infection should be considered when interpreting immune response parameters.
3. Figures presenting the immunocompromised groups don't consistently identify those with autoimmune, hematologic, and solid tumors.
4. Table 1 should also break out the number and distribution of the malignancy cohorts.
5. Specific anti-neoplastic therapies should be shown. The authors did not comment on how antineoplastic agents were removed from blood samples of solid tumor patients prior to T cell response testing or provide documentation that the agents would not interfere with the assay.
6. Figure 1. Statistical analysis is not explained, and legend lacks sufficient explanatory information. For example, the BLQ group shows responses above the T cell response threshold. Was this group compared statistically to the other groups?
7. Figure 2. Composite scores are said to be shown in Figure 2, but this is not diagrammed in the Figure but rather mentioned in the legend, which furthermore does not fully describe what scores were composited.
8. Figure 3. The text refers to statistical differences amongst groups, but this is not shown in the Figure. Also, the legend should include more information regarding the figure and statistics applied. It seems that box and whisker plots were used.
9. Figure 4. Like Figure 3 the legend lacks sufficient explanatory information, and the statistical comparisons are not shown on the graph.
10. The authors refer to their study as a "pilot study." This is not explained. Usually, pilots are for proof of concept or initial hypothesis testing to justify a larger more comprehensive study.
11. The term immunogenic is used improperly, "individuals develop immunogenic responses." This should be "individuals develop immune responses." The antigens of vaccines and organisms that elicit immune responses are immunogenic.

Staff Comments:

Preparing Revision Guidelines

Please return the manuscript within 60 days; if you cannot complete the modification within this time period, please contact me. If you do not wish to modify the manuscript and prefer to submit it to another journal, please notify me of your decision immediately so that the manuscript may be formally withdrawn from consideration by Microbiology Spectrum.

The point of departure of the authors is novel and interesting, they hypothesized that cancer vs non-cancer individuals would mount less robust humoral and/or cellular vaccine-induced immune SARS-CoV-2 responses and conducted the data collection and research. Totally, the draft was well-written, and the statistics collected were clear. However, some detailed problems should have been concerned during the review process.

1. The format of the article should be corrected if possible, and no cover letter have I found in the submitted files. No "purpose, methods, etc." should be listed in the abstract, the authors could benefit from the published articles in "Microbiology Spectrum", and the "Importance" of the article should be added if possible.

2. in the "Methods", the information the authors introduced should be more detailed. What's the basic evaluation criterion to distinguish the "immunocompetent individuals with no underlying disorders" and the "immunocompromised individuals"? In the purpose of the article, the author introduced that cancer patients are at risk for severe COVID-19 outcomes due to impaired immune responses, whether the "immunocompromised individuals" are cancer patients only or have other symptoms? Is there any criterion to distinguish them from healthy individuals? So, I think that's important to introduce more details in this part.

3. What kind of RBD antibodies were tested in this research? Omicron or other types? How are the accuracy and positive rate for testing out? Because the RBD is mutating, it may be better to clarify this.

4. The graphs shown in the draft were not clear enough, please submit the original graphs. Figure legends in the draft should better contain the statistical percentages and total

tested numbers for easier understanding to readers if the authors intend to explain the results. Figure legend 1, the CDI SARS antibody IgG statement was not accurate enough, and I think it may be SARS-CoV-2.

5. It is better to include a statistical analysis of the relationship between the levels of RBD IgG antibodies in the immunocompetent and immunocompromised population in this research who was first infected and have determined the time of infection to explore the duration of antibody.

6. Within the immunocompetent cohort, individuals with previous infection had a higher composite T-cell response (*What is the level?*) than those who were uninfected ($P < 0.05$). In the immunocompromised cohort, the Long-COVID sub-cohort had a higher mean T-cell response (*What is the level?*) than the uninfected ($P < 0.05$) or previously infected ($P < 0.01$) immunocompromised individuals.

7. The statements for all Figures should be more clear and elucidate the significance of the results, not just some statistical illustration without a conclusion, that may weaken the significance of your research.

8. More statistical data should be added if possible, such as mentioned in question 5.

General comment

The study of Titova et al. analyzes SARS-CoV-2 specific antibody and T cell responses in immunocompetent and immunocompromised cohorts. Because the populations analyzed are subject to many variables, the study requires a comprehensive analysis of the influence of those variables. This is not adequately presented. The main findings that immunocompromise is associated with impaired antibody responses and B cell compromise is independent of T cell responsiveness is largely unsurprising, therefore a more detailed presentation of the groups by specific therapy and disease condition would have been of greater value.

Specific comments

1. Group sizes and gender distributions are disparate among groups. The authors recognize this issue in the Discussion but did not attempt to analyze groups by gender or age.
2. The authors refer to “robust” antibody and T cell responses, but do not define what is robust. This should be defined in terms of relationship to resistance to infection or time since antigen exposure. Since antibody levels rise and decay, the time since vaccination or infection should be considered when interpreting immune response parameters.
3. Figures presenting the immunocompromised groups don’t consistently identify those with autoimmune, hematologic, and solid tumors.
4. Table 1 should also break out the number and distribution of the malignancy cohorts.
5. Specific anti-neoplastic therapies should be shown. The authors did not comment on how antineoplastic agents were removed from blood samples of solid tumor patients prior to T cell response testing or provide documentation that the agents would not interfere with the assay.
6. Figure 1. Statistical analysis is not explained, and legend lacks sufficient explanatory information. For example, the BLQ group shows responses above the T cell response threshold. Was this group compared statistically to the other groups?
7. Figure 2. Composite scores are said to be shown in Figure 2, but this is not diagrammed in the Figure but rather mentioned in the legend, which furthermore does not fully describe what scores were composited.
8. Figure 3. The text refers to statistical differences amongst groups, but this is not shown in the Figure. Also, the legend should include more information regarding the figure and statistics applied. It seems that box and whisker plots were used.
9. Figure 4. Like Figure 3 the legend lacks sufficient explanatory information, and the statistical comparisons are not shown on the graph.
10. The authors refer to their study as a “pilot study.” This is not explained. Usually, pilots are for proof of concept or initial hypothesis testing to justify a larger more comprehensive study.
11. The term immunogenic is used improperly, “individuals develop immunogenic responses.” This should be “individuals develop immune responses.” The antigens of vaccines and organisms that elicit immune responses are immunogenic.

October 04, 2023 version 1

Response to Reviewers

Thank you for the thoughtful and thorough review. We have carefully considered the comments and hope that our responses adequately address the issues mentioned.

Reviewer 1

The point of departure of the authors is novel and interesting, they hypothesized that cancer vs non-cancer individuals would mount less robust humoral and/or cellular vaccine-induced immune SARS-CoV-2 responses and conducted the data collection and research. Totally, the draft was well-written, and the statistics collected were clear. However, some detailed problems should have been concerned during the review process.

1. The format of the article should be corrected if possible, and no cover letter have I found in the submitted files. No “purpose, methods, etc.” should be listed in the abstract, the authors could benefit from the published articles in “Microbiology Spectrum”, and the “Importance” of the article should be added if possible.

Response:

A. The cover letter is copied here:

Dear Dr. Christina Cuomo,

We would like to gauge your interest in a new manuscript that addresses SARS-CoV-2 humoral and cellular vaccine-induced immune responses in a total of 594 vaccinated healthy/immunocompetent and immunocompromised individuals. The study was prompted by a desire to better assess the immune status of patients among our cancer host cohort, one of the largest in the NY Metropolitan region.

In this study, we evaluated receptor binding domain (RBD), SARS-CoV-2 spike-protein antibody levels and T cell responses (QuantiFERON SARS-CoV-2) in healthy individuals with no underlying disorders (n=479) and immunocompromised individuals (n=115), including individuals with hematologic malignancies (n=68) and carcinomas (n=12). SARS-CoV-2 vaccinated individuals mounted robust cellular and/or humoral responses, although higher immunogenicity was observed among the immunocompetent compared to immunocompromised populations. The study supports a suppressive role of B-cell targeted therapies in cancer patients on antibody responses, but not on T cell responses to SARS-CoV-2 vaccination. Vaccination was effective in inducing humoral and cellular immune responses in most individuals, a key preventive measure against infection and subsequent severe adverse outcomes.

This study was a collaborative effort involving the Center for Discovery and Innovation (CDI) and John Theurer Cancer Center (JTCC) of Hackensack Meridian Health (HMH), along with Quest Diagnostics. HMH is the largest healthcare system in New Jersey and cared for more than 75,000 COVID-19 patients in its hospitals. The JTCC sees more than 35,000 new cancer patients

a year and performs more than 500 HSCTs. My group has been extensively engaged in COVID-19 studies, publishing 28 papers on diagnostics, therapeutics, immune responses, and host genetics. I also co-lead with Dr. Charles Rice an NIH Center of Excellence in antiviral discovery.

I look forward to your response.

Sincerely,

David S Perlin, PhD

CSO/EVP and Professor

CENTER FOR DISCOVERY AND INNOVATION

111 Ideation Way, Nutley, New Jersey 07110

T: 201.880.3100 | M: 914.260.2473

Email: david.perlin@hmh-CDI.org; Website: <https://hmh-cdi.org/>

Fellow, New York Academy of Sciences

Fellow, American Academy of Microbiology

Professor, Microbiology and Immunology, Georgetown University School of Medicine

B. The Abstract was reformatted as suggested.

Abstract:

Cancer patients are at risk for severe COVID-19 outcomes due to impaired immune responses. However, the immunogenicity of SARS-CoV-2 vaccination is inadequately characterized in this population. We hypothesized that cancer versus non-cancer individuals would mount less robust humoral and/or cellular vaccine-induced immune SARS-CoV-2 responses. Receptor binding domain (RBD) and SARS-CoV-2 spike protein antibody levels and T-cell responses were assessed in healthy individuals with no underlying disorders (n=479) and immunocompromised individuals (n=115). All 594 individuals were vaccinated and of varying COVID-19 statuses (i.e., not known to have been infected, previously infected, or “Long COVID”). We found among immunocompromised individuals, 59% (n=68) had an underlying hematologic malignancy; of those, 46% (n=31) of individuals received cancer treatment <30 days prior to study blood collection. 98% (n=469) of immunocompetent and 81% (n=93) of immunocompromised individuals had elevated RBD antibody titers (>1,000 U/mL), and of these, 60% (n=281) and 44% (n=41) respectively also had elevated T-cell responses. Composite T-cell responses were higher in individuals previously infected with SARS-CoV-2 or those diagnosed with Long-COVID compared to uninfected individuals. T-cell responses varied between immunocompetent vs carcinoma (n=12) cohorts (P<0.01) but not in immunocompetent vs hematologic malignancy cohorts. In summary, most SARS-CoV-2 vaccinated individuals mounted robust cellular and/or humoral responses, though higher immunogenicity was observed among the immunocompetent compared to cancer populations.

Importance: The study suggests B-cell targeted therapies suppress antibody responses, but not T-cell responses, to SARS-CoV-2 vaccination. Thus, vaccination continues to be an effective way to induce humoral and cellular immune responses as a key preventive measure against infection and/or subsequent more severe adverse outcomes.

2. In the “Methods,” the information the authors introduced should be more detailed. What’s the basic evaluation criterion to distinguish the “immunocompetent individuals with no underlying disorders” and the “immunocompromised individuals”? In the purpose of the article, the author introduced that cancer patients are at risk for severe COVID-19 outcomes due to impaired immune responses, whether the “immunocompromised individuals” are cancer patients only or have other symptoms? Is there any criterion to distinguish them from healthy individuals? So, I think that’s important to introduce more details in this part.

Response:

We described, “Immunocompromised participants were self-reported individuals who had an autoimmune disorder or received chemotherapy, immunotherapy, and/or radiation treatments/medications. All other individuals were classified as being immunocompetent.”

3. What kind of RBD antibodies were tested in this research? Omicron or other types? How are the accuracy and positive rate for testing out? Because the RBD is mutating, it may be better to clarify this.

Response: Based on a study conducted within our hospital network (Mediavilla et al.), the predominant variants present during the sample collection time frame were Delta and Omicron (BA.1).

However, T cells recognize an infected cell through any part of the spike protein. The RBD is only a small part of the spike so most T cells are unaffected by changes in the Omicron variants. Dr. Otto Yang, professor, Department of Medicine and Microbiology, Immunology and Molecular Genetics, David Geffen School of Medicine at UCLA, stated, “.. to a T-cell, the omicron variant is still 97% the same as the original strain.” T cells that largely tolerate the amino acid mutations that characterize the different variants of concern, including Omicron.¹⁻⁸

References:

1. De Marco L., D’Orso S., Pirronello M., Verdiani A., Termine A., Fabrizio C., Capone A., Sabatini A., Guerrera G., Placido R., et al. Assessment of T-cell reactivity to the SARS-CoV-2 omicron variant by immunized individuals. *JAMA Netw. Open.* 2022;5:e2210871. doi: 10.1001/jamanetworkopen.2022.10871.
2. Gao Y., Cai C., Grifoni A., Müller T.R., Niessl J., Olofsson A., Humbert M., Hansson L., Österborg A., Bergman P., et al. Ancestral SARS-CoV-2-specific T cells cross-recognize Omicron. *Nat. Med.* 2022;28:472–476. doi: 10.1038/d41591-022-00017-z.
3. Gao Y., Cai C., Grifoni A., Müller T.R., Niessl J., Olofsson A., Humbert M., Hansson L., Österborg A., Bergman P., et al. Ancestral SARS-CoV-2-specific T cells cross-recognize Omicron. *Nat. Med.* 2022;28:472–476. doi: 10.1038/d41591-022-00017-z.
4. Gao Y., Cai C., Grifoni A., Müller T.R., Niessl J., Olofsson A., Humbert M., Hansson L., Österborg A., Bergman P., et al. Ancestral SARS-CoV-2-specific T cells cross-recognize Omicron. *Nat. Med.* 2022;28:472–476. doi: 10.1038/d41591-022-00017-z.

5. Gao Y., Cai C., Grifoni A., Müller T.R., Niessl J., Olofsson A., Humbert M., Hansson L., Österborg A., Bergman P., et al. Ancestral SARS-CoV-2-specific T cells cross-recognize Omicron. *Nat. Med.* 2022;28:472–476. doi: 10.1038/d41591-022-00017-z.
6. Naranbhai V., Nathan A., Kaseke C., Berrios C., Khatri A., Choi S., Getz M.A., Tano-Menka R., Ofoman O., Gayton A., et al. T cell reactivity to the SARS-CoV-2 Omicron variant is preserved in most but not all individuals. *Cell.* 2022;185:1259. doi: 10.1016/j.cell.2022.03.022.
7. Oh B.L.Z., Tan N., Alwis R. de, Kunasegaran K., Chen Z., Poon M., Chan E., Low J.G., Yeoh A.E.J., Bertoletti A., et al. Enhanced BNT162b2 vaccine-induced cellular immunity in anti-CD19 CAR T cell treated patients. *Blood.* 2022 doi: 10.1182/blood.2022016166.
8. Oh B.L.Z., Tan N., Alwis R. de, Kunasegaran K., Chen Z., Poon M., Chan E., Low J.G., Yeoh A.E.J., Bertoletti A., et al. Enhanced BNT162b2 vaccine-induced cellular immunity in anti-CD19 CAR T cell treated patients. *Blood.* 2022 doi: 10.1182/blood.2022016166.
9. Mediavilla, J.R.; Lozy, T.; Lee, A.; Kim, J.; Kan, V.W.; Titova, E.; Amin, A.; Zody, M.C.; Corvelo, A.; Oswald, D.M.; Baldwin, A.; Fennessey, S.; Zuckerman, J.M.; Kirn, T.; Chen, L.; Zhao, Y.; Chow, K.F.; Maniatis, T.; Perlin, D.S.; Kreiswirth, B.N. Molecular and Clinical Epidemiology of SARS-CoV-2 Infection among Vaccinated and Unvaccinated Individuals in a Large Healthcare Organization from New Jersey. *Viruses* 2023, 15, 1699.

4. The graphs shown in the draft were not clear enough, please submit the original graphs. Figure legends in the draft should better contain the statistical percentages and total tested numbers for easier understanding to readers if the authors intend to explain the results. Figure legend 1, the CDI SARS antibody IgG statement was not accurate enough, and I think it may be SARS-CoV-2.

Response:

Graphs have been updated to address the concerns about resolution, clarity in axis titles and expanded legend to explain pertinent details.

5. It is better to include a statistical analysis of the relationship between the levels of RBD IgG antibodies in the immunocompetent and immunocompromised population in this research who was first infected and have determined the time of infection to explore the duration of antibody.

Response:

The relationship between IgG antibodies and elapsed time since infection was explored in the original analysis. Due to the subjectivity of the infection time (patient reported) and lack of trends, it was not shown in manuscript. We are happy to provide this as a supplemental figure.

6. Within the immunocompetent cohort, individuals with previous infection had a higher composite T-cell response (What is the level?) than those who were uninfected ($P < 0.05$) or previously infected ($P < 0.01$) immunocompromised individuals.

Response:

Within the immunocompetent cohort, individuals with previous infection had a higher mean composite T-cell response (0.94 IU/mL) than those who were uninfected (0.57 IU/mL; $P < 0.05$).

7. The statements for all Figures should be more clear and elucidate the significance of the results, not just some statistical illustration without a conclusion, that may weaken the significance of your research.

Response:

Graphs have been updated to address the concerns about resolution, clarity in axis titles and expanded legend to explain pertinent details.

8. More statistical data should be added if possible, such as mentioned in question 5.

Response: Please refer to response to question #5.

Reviewer 2

General comment The study of Titova et al. analyzes SARS-CoV-2 specific antibody and T cell responses in immunocompetent and immunocompromised cohorts. Because the population analyzed are subject to many variables, the study requires a comprehensive analysis of the influence of those variables. This is not adequately presented. The main findings that immunocompromise is associated with impaired antibody responses and B cell compromise is independent of T cell responsiveness is largely unsurprising, therefore a more detailed presentation of the groups by specific therapy and disease condition would have been of greater value.

Specific comments

1. Group sizes and gender distributions are disparate among groups. The authors recognize this issue in the Discussion but did not attempt to analyze groups by gender or age.

Response:

The study aimed to demonstrate SARS-CoV-2 humoral and cellular vaccine-induced immune responses. Although there are differences in immune responses by age and sex, we believe refining the analysis in this way distracts from our overall key observations. We conducted a sensitivity analysis for sex and age, and our initial analysis was robust enough that there were no changes to the significant findings of the original paper.

2. The authors refer to “robust” antibody and T cell responses, but do not define what is robust. This should be defined in terms of relationship to resistance to infection or time since antigen exposure. Since antibody levels rise and decay, the time since vaccination or infection should be considered when interpreting immune response parameters.

Response: The paper’s language was updated to reflect that a robust antibody response was defined as “a detectable result greater than 1,000 U/mL within the HMH-CDI Research Use Only (RUO) SARS-CoV-2 Receptor Binding Domain (RBD) ELISA Assay” and a robust T-cell

response was defined as above the “threshold of 0.2 IU/mL.” We cannot define a robust antibody response based on resistance to infection or time since antigen, especially with self-reported clinical data.

3. Figures presenting the immunocompromised groups don’t consistently identify those with autoimmune, hematologic, and solid tumors.

Response: Due to low sample size within the immunocompromised groups, we felt that stratification in all figures did not elucidate additional insight into the data trends.

4. Table 1 should also break out the number and distribution of the malignancy cohorts.

Response: Thank you for your comment. Further stratification of Table 1 did not elucidate additional insight into the data trends.

5. Specific anti-neoplastic therapies should be shown. The authors did not comment on how antineoplastic agents were removed from blood samples of solid tumor patients prior to T cell response testing or provide documentation that the agents would not interfere with the assay.

Response: There is no evidence to suggest that anti-neoplastic agents will interfere with our assays.

6. Figure 1. Statistical analysis is not explained, and legend lacks sufficient explanatory information. For example, the BLQ group shows responses above the T cell response threshold. Was this group compared statistically to the other groups?

Response: Graphs have been updated to address the concerns about resolution, clarity in axis titles and expanded legend to explain pertinent details. Due to insufficient sample size, pairwise group comparisons were not performed.

7. Figure 2. Composite scores are said to be shown in Figure 2, but this is not diagrammed in the Figure but rather mentioned in the legend, which furthermore does not fully describe what scores were composited.

Response: Graphs have been updated to address the concerns about resolution, clarity in axis titles and expanded legend to explain pertinent details.

8. Figure 3. The text refers to statistical differences amongst groups, but this is not shown in the Figure. Also, the legend should include more information regarding the figure and statistics applied. It seems that box and whisker plots were used.

Response: Graphs have been updated to address the concerns about resolution, clarity in axis titles and expanded legend to explain pertinent details.

9. Figure 4. Like Figure 3 the legend lacks sufficient explanatory information, and the statistical comparisons are not shown on the graph.

Response: Graphs have been updated to address the concerns about resolution, clarity in axis titles and expanded legend to explain pertinent details.

10. The authors refer to their study as a “pilot study.” This is not explained. Usually, pilots are for proof of concept or initial hypothesis testing to justify a larger more comprehensive study.

Response: Thank you for your comments. The purpose of this study was to look for specific trends to establish future directions, as a pilot study would, in order to perform a larger, statistically rounded study in the future. This pilot study was not intended to be a fully powered study, and was more oriented towards data collection as demonstrated by the diverse study population.

11. The term immunogenic is used improperly, “individuals develop immunogenic responses.” This should be “individuals develop immune responses.” The antigens of vaccines and organisms that elicit immune responses are immunogenic.

Response: Thank you. The revisions were made as suggested.

Re: Spectrum02050-23R1 (Humoral and Cellular Immune Responses Against SARS-CoV-2 Post-Vaccination in Immunocompetent and Immunocompromised Cancer Populations)

Dear Ms. Elizabeth Titova:

Thank you for the privilege of reviewing your work. Below you will find my comments, instructions from the Spectrum editorial office, and the reviewer comments (can view under attachments).

Revision Guidelines

Sincerely,
Tulip Jhaveri
Editor
Microbiology Spectrum

General comment: The revised manuscript addresses most of the concerns raised in the initial review. Some minor issues remain.

1. The authors' rebuttal states that there is no evidence that immunosuppressive therapies affect the QuantiFERON assay. A supporting reference should be provided. Studies have shown effects of therapies on the TB version of this assay. For example, steroid therapy can influence the test.
2. The authors have improved the statistical presentation in the revised manuscript. Some questions remain. Figure 1 correlates antibody levels and T cell responses in immunocompetent and immunocompromised groups. It is not clear if a formal correlation analysis was performed to help support the observation. Also, in Figure 2, they state that a pairwise t-test was performed. This test is usually applied to analyze changes in a group before and after an intervention. Was the pairwise t-test recommended by a statistician?
3. The authors address the weaknesses and limitations of the study to some extent in the Discussion section. However, they did not include the fact that the study does not establish the relationship of antibody and T cell responses to clinical protection which would require prospective epidemiological studies.

December 29, 2023 version 2

General comment: The revised manuscript addresses most of the concerns raised in the initial review. Some minor issues remain.

Response to Reviewers: Thank you again for the thoughtful and thorough review.

1. The authors' rebuttal states that there is no evidence that immunosuppressive therapies affect the QuantiFERON assay. A supporting reference should be provided. Studies have shown effects of therapies on the TB version of this assay. For example, steroid therapy can influence the test.

Authors: The manufacturer (Qiagen) has not reported any impact of therapies, including steroids, on test performance. No literature was identified to demonstrate interference from immunosuppressive therapeutics on the assay. In the absence of literature or manufacturers' information, we suggest leaving the statement stand as written.

Reference link to package insert 1095849_R09_QF-TB Gold Plus ELISA IFU_Clean.pdf

Qiagen. QuantiFERON®-TB Gold Plus (QFT®-Plus) Package Insert. January 2023.

2. The authors have improved the statistical presentation in the revised manuscript. Some questions remain. Figure 1 correlates antibody levels and T cell responses in immunocompetent and immunocompromised groups. It is not clear if a formal correlation analysis was performed to help support the observation. Also, in Figure 2, they state that a pairwise t-test was performed. This test is usually applied to analyze changes in a group before and after an intervention. Was the pairwise t-test recommended by a statistician?

Authors: In Figure 1, we investigated an association between antibody and T-cell responses by ANOVA (p 0.64), as the data was non-monotonic. The first three categories were excluded due to low sample size, but the overall results affirmed that there was no difference in the T-cell response between the antibody groups.

For Figure 2, I think a clarification is needed. A "paired" t-test, which is what is appropriate for looking at a change pre/post, was not performed. An ANOVA was run first and then to perform a pairwise group analysis, a two-side independent t-test was used. To reflect this, the Figure 2 legend was updated to reflect that a "pairwise independent t-test" was performed.

3. The authors address the weaknesses and limitations of the study to some extent in the Discussion section. However, they did not include the fact that the study does not establish the relationship of antibody and T cell responses to clinical protection which would require prospective epidemiological studies.

Authors: Thank you for this additional limitation. We added to the Discussion, "Our study, while suggestive, does not firmly establish the relationship of antibody and T cell responses to clinical protection, which would require more extensive prospective epidemiological studies."

Re: Spectrum02050-23R2 (Humoral and Cellular Immune Responses Against SARS-CoV-2 Post-Vaccination in Immunocompetent and Immunocompromised Cancer Populations)

Dear Ms. Elizabeth Titova:

Your manuscript has been accepted, and I am forwarding it to the ASM production staff for publication. Your paper will first be checked to make sure all elements meet the technical requirements. ASM staff will contact you if anything needs to be revised before copyediting and production can begin. Otherwise, you will be notified when your proofs are ready to be viewed.

Sincerely,
Tulip Jhaveri
Editor
Microbiology Spectrum